# Importance of Electrode Selection and Number in Reconstructing Standard Twelve Lead Electrocardiograms

**DOI:** 10.3390/biomedicines11061526

**Published:** 2023-05-25

**Authors:** Adam A. Butchy, Utkars Jain, Michael T. Leasure, Veronica A. Covalesky, Gary S. Mintz

**Affiliations:** 1Heart Input Output Inc., DBA HEARTio, Pittsburgh, PA 15213, USAmichael.leasure@heartio.ai (M.T.L.); 2Cardiology Consultants of Philadelphia, Philadelphia, PA 19148, USA; 3Jefferson University Hospital, Philadelphia, PA 19107, USA; 4The Cardiovascular Research Foundation, New York, NY 10019, USA

**Keywords:** 12-lead reconstruction, lead placement, lead importance, lead significance, artificial intelligence

## Abstract

Many clinical and consumer electrocardiogram (ECG) devices collect fewer electrodes than the standard twelve-lead ECG and either report less information or employ algorithms to reconstruct a full twelve-lead signal. We assessed the optimal electrode selection and number that minimizes redundant information collection while maximizing reconstruction accuracy. We employed a validated deep learning model to reconstruct ECG signals from 250 different patients in the PTB database. Different numbers and combinations of electrodes were removed from the ECG before reconstruction to measure the effect of electrode inclusion on reconstruction accuracy. The Left Leg (LL) electrode registered the largest drop in average reconstruction accuracy, from an $R^{2}$ of 0.836 when the LL was included to 0.737 when excluded. Additionally, we conducted a correlation analysis to identify leads that behave similarly. We demonstrate that there exists a high correlation between leads I, II, aVL, aVF, V4, V5, and V6, which all occupy the bottom right quadrant in an ECG axis interpretation, and likely contain redundant information. Based on our analysis, we recommend the prioritization of electrodes RA, LA, LL, and V3 in any future lead collection devices, as they appear most important for full ECG reconstruction.

## 1. Introduction

Electrocardiograms (ECGs) are a common diagnostic tool used to evaluate the electrical activity of the heart. There are three main types of ECGs, which differ in electrode number and clinical use: the standard twelve-lead ECGs, which are often used in a clinical setting to detect myocardial infarction, ischemia, or arrhythmias [1,2]; the three- or five-lead Holter monitors, which are used at the behest of a clinician to catch more sporadic arrhythmias and for longer monitoring [3]; and the one-lead rhythm monitors (usually in the form of patches or smartwatches), which are used as at home devices or consumer products [4,5]. The number and placement of electrodes differ for each type, thus affecting the quality and amount of information that can be obtained. 

Standard twelve-lead ECGs employ ten electrodes placed on the patient’s chest and limbs to generate the six limb leads (I, II, III, aVF, aVL, and aVR) and six chest leads (V1–V6) depicted in Figure 1. The electrodes are placed strategically to create a three-dimensional view of the heart’s electrical activity [6]. Other lead configurations exist for different clinical settings, enhanced atrial activity detection, or improved visualization of the Epsilon wave, and they have been reviewed elsewhere [7]. Lead configurations must balance clinical utility, patient comfort, and signal collection and quality. In the case of fewer-than-twelve-lead devices—i.e., Holter Monitors or patches that employ one or three or five electrodes—algorithms are often employed that try to extrapolate or reconstruct a full twelve-lead ECG from the leads that are collected. The most common approaches to reconstruct a full twelve-lead ECG are via complex linear regressions [8], principal component analyses [9], or artificial intelligence [10]. 

When a manufacturer designs a new ECG device, they must consider electrode placement and lead configuration in order to balance clinical utility, patient comfort, and signal collection and quality. In a previous study [11], we examined the importance of single leads in reconstructing a standard twelve-lead ECG. In this paper, we examine the importance of electrode placement and lead selection—alone and in various combinations—for the purpose of full ECG reconstruction. Primarily, we aim to identify the electrodes and leads that contribute to the most faithful ECG reconstructions. Secondarily, we conduct a correlation analysis to determine which leads behave most similarly (pointing at the fact they may contain redundant information) and discuss why they correlate with each other. 

## 2. Materials and Methods

### 2.1. The Deep Learning Model

In a prior work [11], we validated a deep learning model (ECGio version 16.0, Heart Input Output Inc, Pittsburgh, PA, USA) and its ability to reconstruct missing ECG information for 250 patients randomly selected from the PTB database [12]. This model was created to use the first 10 s from a patient’s ECG as input to classify the signal according to a variety of clinically relevant outputs including, but not limited to, abnormal beats and rhythms, the presence of myocardial infarction, and the patient’s severity of coronary artery disease [13]. For more information on the deep learning model, please refer to the previous work in [11,13,14]. 

For this current study, the model was trained through an iterative process known as backpropagation (BP), in which data is fed into a feedforward network (i.e., the data flows from the head of the model to the tail of the model), and the difference between the intended output and the predicted output or “loss” is calculated. This loss is propagated back through the network as weights, or parameters, of the model and is adjusted based on contributions to the loss. This training process is simplified and depicted in Figure 2. One pass of data through the model is known as a “forward pass”—passing the data in the forward direction. The passing of the loss through the model from the tail to the head is known as a “backward pass”—adjusting weights while passing the loss through the opposite direction of the model.

Training the AI model is only the first step in the process of successfully creating a useful model. The model must provide meaningful results and not simply memorize the training data. Verification of a useful model can be achieved by splitting the dataset into three specific parts, where one part is used for training the model to optimize the weights through BP (the “training set”), one is used to validate that the model is learning useful parameters (the “validation set”), and the last one is used to test the model in a completely blind fashion (the “test set”). Other methods for verifying model utility exist, such as the “K-fold cross-validation” and the “leave-one-out” methods, which provide additional insights regarding the performance of the model [15]. In this study, we utilized only a training set and a test set. The previously trained model was not privy to any of the signals found in the test set, and the test set was not used to validate model performance. The model was trained on clinical labels and ECG reconstruction, which, based on the size and architecture of the model, represented too much information for the model to simply memorize. This limitation forced the model to distill and correlate information, making for a more generalized model, rather than one that memorizes the ECG data explicitly.

When the model is asked to “fill in” missing lead information, it is assumed that missing lead information is zeroed out. The model then assumes that this portion of the ECG cannot be used for the purposes of lead reconstruction. 

### 2.2. The PTB Database and Data Cleaning

We used the large open-source Physiobank’s Pysikalisch-Technische Bundensanstalt (PTB) database of diagnostic ECG created by the National Metrology Institute of Germany. Patients both with and without cardiovascular disease had their ECG recorded at the Department of Cardiology of the University Clinic Benjamin Franklin in Berlin Germany. The database consists of 290 patients. Patient demographics are summarized below in Table 1.

Both patient-specific and environmental factors can introduce noise into the ECG signal (body habitus, hair, sweat, muscle movement, electrical interference, among others) [16]. ECGs typically undergo many different types of signal processing to mitigate the patient-to-patient and environment-to-environment variations in ECG signals, and to account for the different sources and magnitude of collection noise. It is important to have a preprocessing standardization technique to make sure there is level comparison for both the training and testing of the model. We do this standardization in two ways: (1) filtering the signal to isolate specific frequency bands of the ECG and (2) normalizing the amplitude. The signals are filtered with a bandpass Butterworth filter at a passband from 2–40 s^−1^ and normalized to unitless values between −1 and +1. A bandpass filter is used below 40 s^−1^, where most of the higher-frequency components are thought to occur [17] and above 2 s^−1^, where most of the low-frequency components are thought to occur [18]. A Butterworth filter is used, as it is the smoothest of the major filter types (e.g., Butterworth, Chebyshev, Bessel) with no ripples in the frequency domain—which is hypothesized to be important for retaining the original information of the signal. In simpler terms, we sacrifice a narrower transition band for a smoother filtering experience.

### 2.3. Lead Combinatorial Analysis

To determine which leads were most important for a limited lead set, different leads were systematically included or removed from the same 250-patient ECG set. All possible combinations of lead inclusion were explored for each patient. In this situation, we let S = [1, 2, 3, …, 11]; S_r_ represents a set of all the combinations of S with r length combinations; and A represents the set that is the union of all the combinations, A = [S1, S2, S3, …, S10, S11]. Each set in A indicated which leads in the input ECG should remain intact, and all others were masked to zero. This process is depicted in Figure 3A. Performance statistics were averaged across each S_r_. As per the pattern of combinatorics, there were a lot more combinations for six leads (924 combinations) than for one lead (12 combinations). We included the equation that describes the number of lead combinations (S_r_) based on the number of leads used (r) in Figure 3B. Each of the 250 patients were used to create a total of 4094 different ECGs (with varying numbers of leads), resulting in a dataset of 1,023,500 different ECGs.

### 2.4. Lead Correlation Analysis

To quantify the amount of unique information in each ECG lead, we correlated each patient’s twelve leads to each of their other leads within the same ECG signal. This correlation coefficient quantified the degree that two leads behaved similarly. Leads that perfectly matched each other registered a correlation coefficient of 1. Leads that were mirror reflections had a correlation coefficient of −1. Leads that behaved completely independently had a correlation coefficient of 0. We took the absolute value of these correlation coefficients and took the mean and standard deviation values for each lead as it was compared against all the others. This resulted in a 12 × 12 matrix in which each cell represents the mean correlation coefficient for each ECG lead against the other ECG leads. Cells on the diagonal have a value of 1, as each lead, when compared against itself, has a perfect correlation.

### 2.5. Electrode Combinatorial Analysis 

To determine which electrodes were most important to collect in a fewer-electrode device, different electrodes needed to be systematically included or removed. Although the optimal way to do an electrode combinatoric analysis would be to use data of specific electrodes (RA [right arm], LA [left arm], RL [right leg], etc.), this information was not available in this 250 patient dataset; furthermore, the AI model was not capable of using just electrode data from the limb electrodes to reconstruct ECGs. Based on our previous analysis, it was decided that a good proxy for a full electrode combinatoric analysis was to insert which electrodes were present when a particular lead was present; for example, if lead I was used to reconstruct the ECG signal, both the RA and LA electrodes were used.

### 2.6. The Signal-to-Noise Ratio Calculation

In signals subject to noisy collection, a common metric to quantitatively describe the clarity of the signal is the Signal-to-Noise ratio (SNR). The SNR is defined as the ratio of the signal’s mean to the signal’s standard deviation [19].
(1)$$SNR\left(x\right)=\frac{\bar{x}}{\sigma_{x}}$$
where, $\bar{x}$ is the sample mean and $\sigma_{x}$ is the sample standard deviation. Since the ECG’s signal average may be negative, we define $\left|SNR\right|$ to preserve the ratio of signal-to-noise, but make all values positive.
(2)$$\left|SNR\right|\left(x\right)=\left|\frac{\bar{x}}{\sigma_{x}}\right|$$

An $\left|SNR\right|$ was calculated for each of the 1,023,500 signals used in this analysis, and this metric served as an approximate but objective means to quantify the noise in the ECG signal.

### 2.7. Statistics and Error Calculations

To assess signal reconstruction proficiency, we calculated the mean absolute error (MAE) and Pearson correlation coefficient ($R^{2}$) to quantify the difference between the true patient ECG and the results of reconstruction. The MAE provides an indication of the point-to-point distance between the original ECG and the reconstructed ECG.
(3)$$MAE\left(y,\hat{y}\right)=\frac{1}{N}\sum_{i=1}^{N}\left|y_{i}-\hat{y_{i}}\right|$$
$R^{2}$ quantifies the strength of the linear relationship between the original ECG and the reconstructed ECG (i.e., if one went up, did the other go up?).
(4)$$R^{2}\left(y,\hat{y}\right)=\frac{E [\left(y-\mu_{y}\right)\left(\hat{y}-\mu_{\hat{y}}\right)]}{\sigma_{y}\sigma_{\hat{y}}}$$

Both metrics are commonly used in assessing reconstruction accuracy [20]. In isolation, these metrics give a limited understanding of the performance of the model. A model can have a low MAE value but a low reconstruction performance if the amplitude of the of original ECG was low and the model selected a mean value for signal reconstruction [21]. On the other hand, a model can have a high correlation coefficient but low reconstruction performance if the model generally matches the variance of the ECG, but deviates from the original signal to provide a meaningful reconstruction [22]. By using both of these different metrics, we make sure that that the model is providing a meaningful reconstruction. 

## 3. Results

### 3.1. Lead Importance in Fewer Lead Reconstructions

The majority of reconstructions were highly accurate, with an average MAE of 0.054 and an average $R^{2}$ of 0.834. Looking specifically at lead reconstruction using fewer leads, we showed that the greater the number of leads used for reconstruction, the better the lead reconstruction as calculated by MAE and $R^{2}$. In Figure 4, we looked at the top 10% of reconstructions in each group of number of leads. For each group (e.g., two leads used in reconstruction), we calculated the frequency of lead appearances in the best reconstructions. Leads that appeared more frequently than random chance were depicted as positive, while those appearing less frequently than random chance appeared as negative. This discrepancy showed which leads were useful for reconstruction.

Please note that limb leads (specifically leads I, II, aVL, and aVF) were included most often when there were fewer leads used in reconstruction. As more leads were added, chest leads became more important; and limb leads were less often included. This behavior pointed to limb leads encapsulating a lot of important information by themselves, making them more likely to be used when the AI model only included a few leads. As the AI had access to more leads, limb leads lacked the granular information available from more specific chest leads. 

### 3.2. Lead Similarity and Correlation

To explore which leads carried similar information, we analyzed how strongly leads correlated with each other in the same patient (this methodology is explained in Section “Lead Correlation Analysis”). We calculated cross-correlation matrices to evaluate the level of linear relationships between different ECG leads within the same patient. Values that were closer to 1 indicated that the two leads shared a positive linear relationship (i.e., both leads varied similarly). Values closer to −1 indicated that the leads shared a negative linear relationship (i.e., leads varied oppositely). Values close to 0 indicated an uncorrelated relationship. We averaged lead correlations across the 250 patients and created the 12 × 12 matrix shown in Figure 5.

From Figure 5, we see that leads III, V2, and V3 were the most linearly uncorrelated to the other nine leads; but leads V2 and V3 were more similar to each other than when compared to the other 10 leads. This is captured in the fact that the rows and columns of leads III, V2, and V3 are the most consistently dark blue. This supported the findings in Figure 4, as leads I, II, aVL, and aVF were strongly correlated, but somewhat surprisingly, so were leads V4, V5, and V6. In fact, leads V4, V5, and V6 were perhaps the most strongly positively correlated, with the correlation between leads V5 and V6 being so close that it rounded to 1.00. 

From an AI perspective, it may be redundant to have multiple leads that have a very high positive or negative relationship (close to 1 or −1) and beneficial to pick ECG leads that have an uncorrelated relationship (close to 0) to incorporate novel, uncorrelated information. This could be a possible explanation as to why chest leads became more valuable as more leads were allowed; the correlation between the limb leads (specifically leads I and II) was much higher than the correlation between lead I and V2 or V3.

### 3.3. Electrode Importance in Fewer Lead Reconstructions

When considering which electrodes led to the best reconstruction, we expanded our view to the top and bottom 20% of reconstructions. For each reconstruction, we determined which specific electrodes had been used for a particular reconstruction. We then calculated the frequency of their usage. The results of this analysis are shown in Figure 6.

The most important electrodes used for two-electrode reconstructions were lead V5 and lead V6, with the least important being leads V1 and V2. In the three-electrode configurations, V2 and V3 gained significance, while V1 and V4 remained irrelevant. Conversely, V2 also appeared frequently in the bottom 20% of the three-electrode configuration. This contradiction may hint at the information content of V2 being important, but contextually inconsistent or misleading. In the three- to eight-electrode configurations, limb leads decreased in importance. We quantified the differences in performances between electrode configurations and reported the results in Table 2. Only the RA, LA, LL, and V3 electrodes contribute significantly to reconstruction performance.

### 3.4. Effect of the Signal-to-Noise Ratio on Reconstruction Performance

We randomly selected 250 patients out of the 290 in the PTB database, with no signals being excluded for noise or artifacts. We did not use the entire database due to the combinatoric nature of our analyses and our computational restrictions. Since we did not account for noisy signals or those that may contain artifacts, we were interested in seeing if noisy signals—as determined by their Signal-to-Noise ratio (SNR)—influenced reconstruction performance. We calculate a signal’s SNR for each of the 1,023,500 signal combinations used in the analyses above. The absolute value of the signal’s SNR’s is plotted it in Figure 7A. To examine the effect of $\left|SNR\right|$ on reconstruction performance, we created Figure 7B by binning signals based on their $\left|SNR\right|$ and identifying outliers in each bin. 

We observed a small but steady decrease in MAE (denoting an increase in reconstruction performance) in signals with higher $\left|SNR\right|$. When the average MAE was calculated for signals with different binned values of $\left|SNR\right|$, this trend was preserved. This trend was preserved despite the presence of outliers, which occurred at many different levels of SNR, and yet resulted in much worse reconstruction performance. 

## 4. Discussion

Our analysis reveals the different levels and quality of information present within each ECG lead and electrode, and their importance to standard twelve-lead reconstruction. The techniques described herein identify which leads carry unique information and contribute to reconstruction accuracy. In designing fewer-electrode ECGs, we must not only think about which electrodes are easiest to collect, but which will return the most beneficial information with which to reconstruct the full twelve-lead signal.

Our correlation analysis (Figure 5) presented an interesting finding with the high correlation between leads I, II, aVL, aVF, V4, V5, and V6, and the likelihood of redundant information therein. Viewed through a lens of ECG axis interpretation, lead I, II, aVF, V4, V5, and V6 all occupy the bottom right quadrant, grouped on the side of the left ventricle (and lead aVL is only slightly above this quadrant). While leads V2 and V3 are on the edge of the same quadrant, they are placed closer to the heart and more towards its front than its side. Considering the anatomy of the heart, these leads (I, II, aVL, aVF, V4, V5, and V6) are on the opposite side of the heart from the sinoatrial and atrioventricular nodes. This anatomic geography contextualizes why these leads appear irrelevant to AI-enabled reconstruction; however, further investigation is warranted. 

Interestingly, the limb leads had high importance in Figure 4, but the left leg electrode appeared frequently in the bottom 20% of reconstructions in Figure 5. The left leg electrode is necessary for calculating leads I, III, aVR, aVL, and aVF but is typically placed the farthest from the heart. The high appearance in bad reconstructions was likely due to the relatively larger amount of noise in this electrode, and noisy collection resulting in poor reconstructions.

Our analysis indicates the presence of a dichotomy, specifically pertaining to the inclusion of the limb leads. If only a limited number of leads are to be collected (e.g., to maximize patient comfort or hardware-based device constraints), there seems to be a greater benefit in using limb leads rather than precordial (chest) leads through the top 20% of performance reconstructions. This is consistent with lead placement of popular three-lead Holter monitors (utilizing left arm, right arm, and left leg leads). However, once more leads are collected, our analysis points to the value of chest leads, focusing on V1-3, to ensure high reconstruction fidelity. This observation warrants further study; however, our leading hypothesis is that the limb leads provide broad, coarse information that is useful for the majority of the reconstruction details (e.g., QRS location and orientation), while chest leads provide more granular, clarifying information (e.g., peak height or effective refractory period). 

As should be expected, reconstruction performance always increases with more electrodes, leads, and information. The question falls to practicing cardiologists on the use cases and utility of reconstructions, as the best possible reconstruction is, at best, a well-collected standard twelve-lead ECG. This is not to say reconstruction algorithms are only for fewer-lead devices. Even when all twelve leads are being collected, reconstruction algorithms can be useful to cardiologists if one or more leads are noisy, misplaced, or poorly attached.

When analyzing if the quality of the ECG signal affected reconstruction performance, we did observe a steady increase in reconstruction performance (as seen in a decrease in MAE) with increasing $\left|SNR\right|$. In signals with $\left|SNR\right|$ close to 0 (meaning low signal to noise in the sample), we observed higher MAE values. This is what one would expect, as noisy signals would be harder for an AI model to reconstruct. What we did not expect was for the relationship to be so mild. There is only a slight downward trend in MAE with increased $\left|SNR\right|$. It appears the majority of higher MAE reconstructions are outliers and that our method is largely robust to low $\left|SNR\right|$. Somewhat unexpectedly, the highest $\left|SNR\right|$ signals have higher average MAE. We attribute this mostly to the low number of signals at this $\left|SNR\right|$. The large presence of outliers represents an area of potential work. In future work, we aim to artificially add noise to clean ECGs, and to control the presence and level of noise in an ECG to observe the impact that artificial noise has on reconstruction performance. 

This analysis represents early work in a quickly growing field. With the proliferation of more affordable and accessible consumer ECGs, and with computational resources getting cheaper and more powerful, we are just beginning to see consumer products used for clinical utility. Whether these devices will be used to automate the detection of transient arrhythmias [23,24] or will only be used to capture signals and send them to a cardiologist for review remains unclear. Our analysis is meant to guide the design and implementation of hardware and software in the ECG space such that high-value leads are prioritized, resulting in the highest quality signal being collected and used for the wellbeing of the patient.

The study was conducted with some limitations. The effect of abnormal beats and rhythms on reconstruction was not examined. The mean absolute error (MAE) and Pearson correlation coefficient ($R^{2}$) cannot quantify the clinical utility of a reconstruction or whether diagnostic information is preserved in the reconstruction. The sample size (250 patients) is comparatively small and further study is warranted to see if these findings are generalizable to a larger, more general population.

## 5. Conclusions

For ECGs collected using fewer than ten electrodes and the standard twelve leads, reconstruction remains a practical solution to outputting a full standard ECG signal. We would advise the prioritization of leads RA, LA, LL, and V3 in any fewer-lead collection devices, as they appear more important for recreating a full twelve-lead system. Additionally, more work remains to determine if the high correlation observed between leads I, II, aVL, aVF, V4, V5, and V6 (which all occupy the bottom right quadrant in an ECG axis interpretation) truly contain redundant information, and can therefore be condensed into fewer electrodes in future devices. 

## Figures and Tables

**Figure 1 biomedicines-11-01526-f001:**
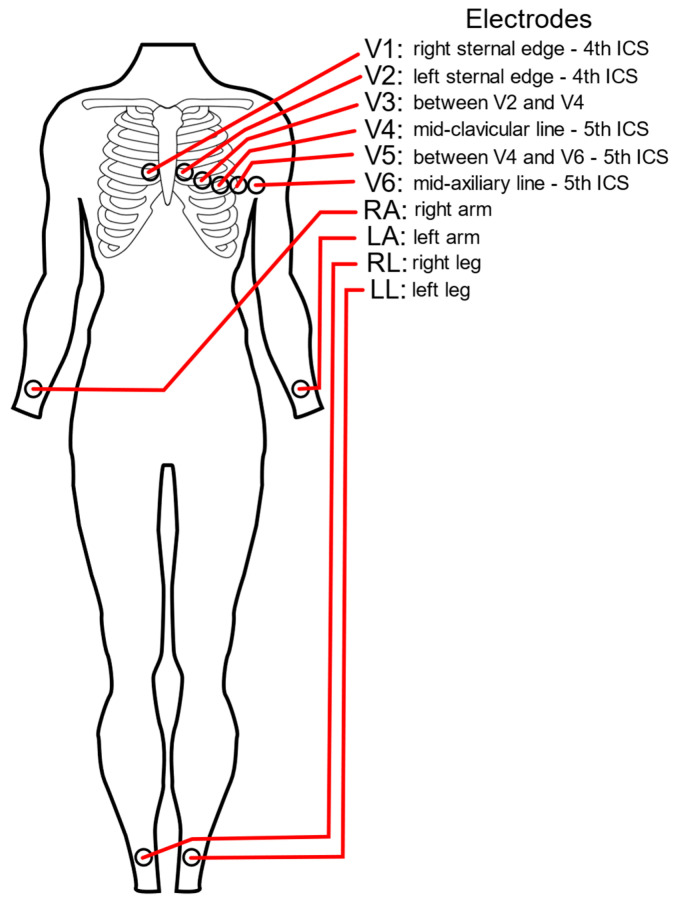
ECG electrode placement: The name and placement of the standard ten electrodes used to collect and calculate a standard twelve-lead ECG. These ten electrodes include the limb leads (RA—right arm, LA—left arm, RL—right leg, and LL—left leg) as well as the chest leads (V1–V6). Typically, the RL lead is used as a ground electrode.

**Figure 2 biomedicines-11-01526-f002:**
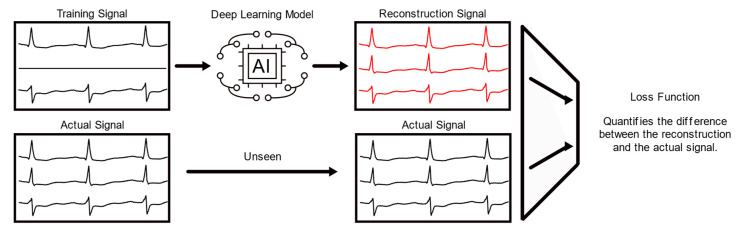
Simplified AI reconstruction: Simplified example of model training in which artificial intelligence attempts to fill in missing leads. The loss function quantifies the difference or “error” between the model’s reconstruction and the actual patient signal. This quantified error informs the model about how faithfully it reconstructed the signal.

**Figure 3 biomedicines-11-01526-f003:**
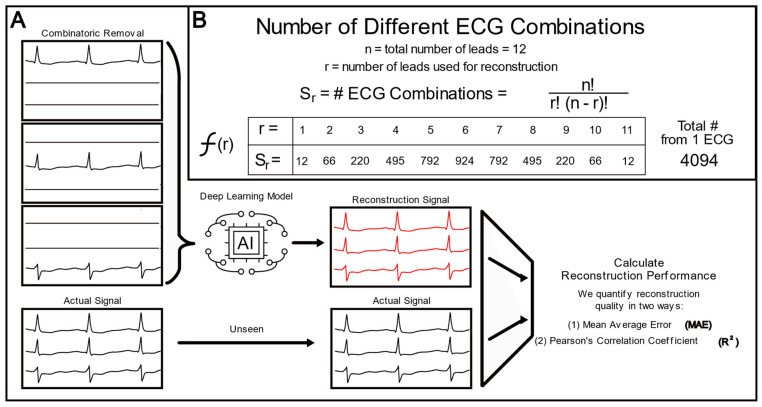
Combinatoric ECG Reconstruction: (**A**) Leads were removed from the same signal in a systematic, combinatoric process, and the model was tasked with reconstructing the ECG based on the remaining leads. The reconstructions were compared against the actual signal to calculate performance metrics. (**B**) The equation that governs the combinatoric inclusion and exclusion of ECG lead information, explicitly defining the number of missing lead ECGs created from just one patient ECG.

**Figure 4 biomedicines-11-01526-f004:**
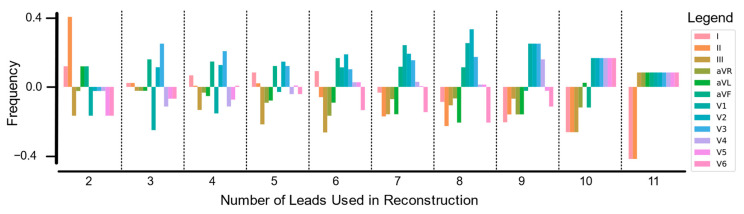
Lead Frequency in top 10% of Reconstructions: The top 10% of reconstructions for each of the number of lead combinations. For each of the best reconstructions, the specific leads used to reconstruct the signal were tallied and reported based on how frequently they were used. This was normalized based on how frequently each lead would be chosen if reconstruction performance was randomized (represented by 0 on the *y*-axis). Leads that appear more frequently than chance are shown as positive frequencies while leads chosen less frequently appear as negative.

**Figure 5 biomedicines-11-01526-f005:**
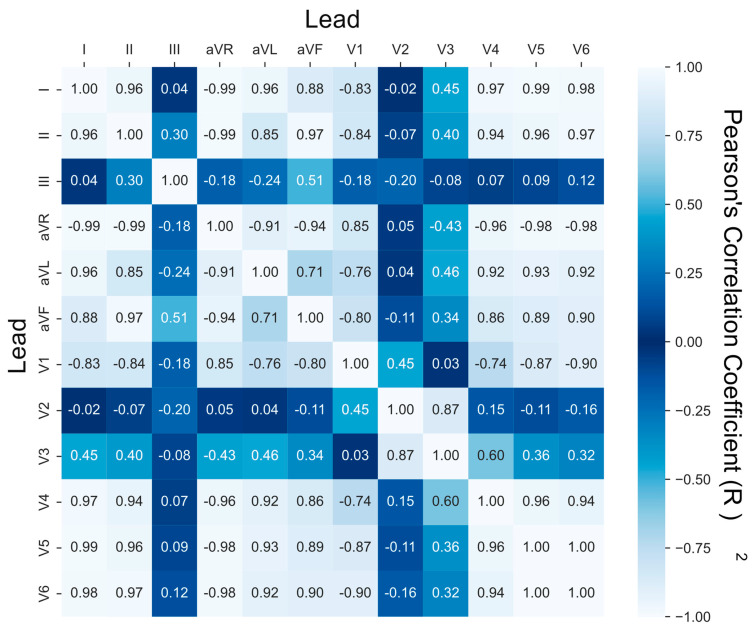
Lead Cross-Correlation Matrix: The cross-correlation matrix quantifying the extent to which two leads were linearly correlated. A value of 1 denotes perfect linear correlation, −1 perfect inverse correlation, and 0 no correlation. Values close to 0 are depicted as darker blue. Values on the diagonal (when each lead was compared against itself) all equal 1. This matrix is a Hankel matrix—symmetric when reflected across the descending left-to-right diagonal.

**Figure 6 biomedicines-11-01526-f006:**
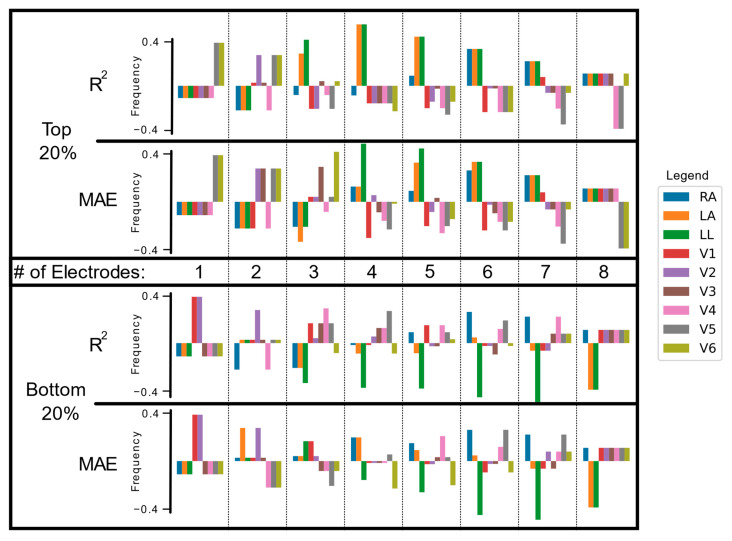
Electrode Frequency in the Top and Bottom 20% of Reconstructions: The electrode frequencies shown in the top and bottom 20% of reconstructions. For each of the best reconstructions, the specific electrodes used to reconstruct the signal were tallied and reported based on how frequently they were used. This frequency was normalized based on how frequently the electrode would be chosen if reconstruction performance was randomized (represented by 0 on the *y*-axis). Electrodes that appeared more frequently than chance are shown as positive frequencies while those chosen less frequently appear as negative.

**Figure 7 biomedicines-11-01526-f007:**
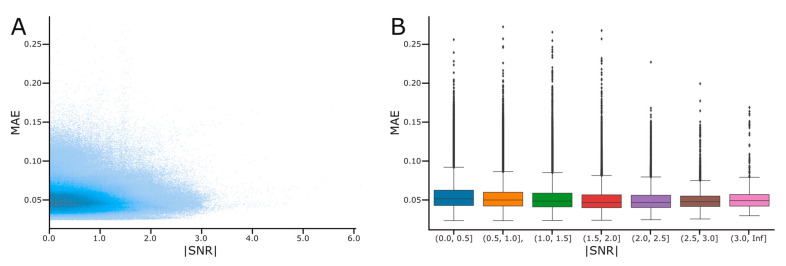
Absolute Signal-to-Noise Ratio ($\left|SNR\right|$ for each Signal: (**A**) A scatterplot of the absolute value of SNR for each of the 1,023,500 signals used in this analysis. (**B**) A box-and-whisker plot for each signal binned based on its |SNR|. Outliers are visualized as black diamonds.

**Table 1 biomedicines-11-01526-t001:** The sex and age demographics of the PTB Database.

	N (%)	Age
Male	209 (72.1%)	55.5
Female	81 (27.9%)	61.6
Total	290 (100%)	57.2

**Table 2 biomedicines-11-01526-t002:** Electrode Significance on Reconstruction: The Pearson correlation coefficient ($R^{2}$) of reconstructions based on which electrodes were used. For each reconstruction, we tallied the leads that were used and the resulting $R^{2}$ value of the reconstruction. Looking at each lead, we calculated the average $R^{2}$ value for reconstructions that used that lead (shown in the “w/lead column”) and compared it to the average $R^{2}$ value of reconstructions that did not use that electrode (shown in the “w/o lead column”). Based on the reconstruction average performance with and without the lead, we calculated if the lead’s inclusion was statistically significant (*p* < 0.05).

Electrode	Reconstruction		*p*-Value
	w/Lead	w/o Lead	
	$R^{2}$	$R^{2}$	
RA	0.836	0.754	0.000
LA	0.836	0.743	0.000
LL	0.836	0.737	0.000
V1	0.841	0.826	0.157
V2	0.843	0.824	0.059
V3	0.844	0.823	0.049
V4	0.841	0.825	0.112
V5	0.839	0.827	0.188
V6	0.839	0.828	0.261

## Data Availability

The data presented in this study may be available on request from the corresponding author. The data are not publicly available due to patient protected health information.

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
