# Peer review of "Importance of Electrode Selection and Number in Reconstructing Standard Twelve Lead Electrocardiograms"

_biomedicines, 2023, doi:10.3390/biomedicines11061526_

Round 1

Reviewer 1 Report

The manuscript by Butchy is very interesting, but it presents some issues that have to be solved. Following, some comments:

                   Format of the paper should be revised: (1) “ie” should be substituted with “i.e.”; (2) units should be reported according to SI; (3) format does not fit the journal guidelines.

                   Abstract should contain background and quantitative results. I suggest revising the abstract.

                   Figure one should be clarified. This is not the standard placement of 10 electrodes for the standard 12-lead ECG recording neither for the standard configuration neither for the Mason-Likar configuration. I strongly suggest correcting the figure.

                   Database should be presented in a separate section. Moreover, which are the anatomical features of these patients? Which is the technical setting of acquisition of the considered ECG?

                   About the preprocessing, why did the authors select a cut-frequency of 2Hz? It is not used in standard ECG signal processing, and this can delete all low-frequency content of the signal. Moreover, which is the order of the filter? Did the author apply a bidirectional filtering?

                   Statistics should be described in a separate section.

                   Did the authors evaluate the dependency of the method to signal to noise ratio and sample frequency of the recording? In my opinion, these features may interfere with the results. I suggest inserting a robustness analysis to these features by using a simulated dataset.

                   I suggest considering real clinical scenario that may benefit from this technology. One can be the use of these algorithms in case of portable and wearable devices (e.g., DOI 10.1109/TCE.2023.3237715; DOI 10.3390/s20123570; DOI 10.1007/978-981-19-5221-0_5). I suggest at least discussing this scenario in the discussion section.

Reviewer 2 Report

This article focuses on the use of less than twelve leads for reconstruction of the classic two-lead ECG. The work is interesting, but I'm not sure about it's novelty to the research group. 

1) Try shortening the title of the article. A good and punchy title has less than eight words without prepositions.

2) The images could be of better quality and preferably vector.

3) I would expect more description of the neural network used.

4) I miss novelties at work. In the introduction of the thesis, clearly state the novelty.

5) From the results of the paper, I don't think the reader is able to reproduce your experiment. I would have expected a better description and demonstration.

Round 2

Reviewer 1 Report

The manuscript still presents some issues that need be solved:

- Units are still not in SI format.

- As authors reported in the answer to reviewers, the SNR interferes with the results. I strongly suggest including the robustness analysis in this paper and not only in the answer to Reviewers.

- In my opinion, clinical utility of the proposed method in the real clinical scenario should be discussed in more detail.

Reviewer 2 Report

The authors have corrected the article according to my recommendations. I now recommend it for acceptance.

Author Response

Reviewer recommended for acceptance

Round 3

Reviewer 2 Report

The authors have corrected the article according to my recommendations and those of other reviewers. I now recommend it for acceptance.

Author Response

No Comments Provided